# The Contribution of Lipotoxicity to Diabetic Kidney Disease

**DOI:** 10.3390/cells11203236

**Published:** 2022-10-14

**Authors:** Jeffrey R. Schelling

**Affiliations:** Department of Physiology and Biophysics, School of Medicine, Case Western Reserve University, Cleveland, OH 44106-4921, USA; jrs15@case.edu; Tel.: +1-216-368-1100

**Keywords:** cholesterol esters, diabetic kidney disease, fatty acids, lipid droplets, podocyte, proximal tubule

## Abstract

Lipotoxicity is a fundamental pathophysiologic mechanism in diabetes and non-alcoholic fatty liver disease and is now increasingly recognized in diabetic kidney disease (DKD) pathogenesis. This review highlights lipotoxicity pathways in the podocyte and proximal tubule cell, which are arguably the two most critical sites in the nephron for DKD. The discussion focuses on membrane transporters and lipid droplets, which represent potential therapeutic targets, as well as current and developing pharmacologic approaches to reduce renal lipotoxicity.

## 1. Lipotoxicity Overview

Lipotoxicity is a metabolic condition that results from the intracellular accumulation of toxic lipid intermediates in non-adipose tissue, leading to cellular dysfunction and potentially cell death (lipoapoptosis) [1]. The prevalence of lipotoxicity is estimated to be 25% in adult populations [2]. The most common affected tissues are kidney, liver (leading to hepatosteatosis), heart (leading to congestive failure), pancreas (leading to β-cell death and diabetes), and skeletal muscle [3]. The parallels between fatty liver disease and diabetic kidney disease (DKD) are increasingly recognized [4]. In liver, there is a well-described spectrum from hepatosteatosis [lipid droplet (LD) accumulation, no inflammation or fibrosis) to steatohepatitis [hepatocyte ballooning, smaller LDs, increased toxic fatty acid (FA) metabolites, no fibrosis] to cirrhosis (fibrosis, fewer LDs) [5,6]. A similar scenario is likely to be apt with DKD, wherein LDs are associated with cytoprotection, and toxic FA metabolites lead to tubular atrophy, interstitial fibrosis and DKD progression.

In kidney, the association between renal tubular epithelial cell lipid deposition and DKD was described many years ago [7,8], and lipotoxicity has been implicated in chronic kidney disease (CKD) progression in humans and animal models [9,10,11,12,13,14]. Nevertheless, the impact of lipotoxicity in human DKD pathogenesis continues to be underestimated because clinical renal biopsies are typically not performed in diabetic people, and biopsy tissue is not routinely processed for lipid detection.

Diabetes is a lipogenic state and the toxic effects of excess lipids are buffered by uptake of FAs by adipocytes. However, when lipid storage capacity is exceeded, FAs are released to the circulation, and either metabolized by the liver and/or stored ectopically in other organs. Lipotoxicity has been associated primarily with ramifications of long-chain (C12-C22) FA metabolite accumulation, and cytotoxicity is proportional to acyl chain length and carbon bond saturation [15,16]. This is relevant to diabetic complications, since plasma concentrations of saturated FAs palmitate (C16:0) and stearate (C18:0) are increased in diabetes [17], and correlate with CKD stage [18]. Palmitate is particularly noxious, since it does not readily incorporate into triacylglycerol (TAG) or cytoprotective LDs, and is instead shunted to ceramide formation in the endoplasmic reticulum (ER) or stimulation of reactive oxygen species (ROS) generation [19], both of which lead to apoptosis [20,21,22,23]. Some, but not all studies indicate that unsaturated FA exposure may be cytoprotective [24,25], perhaps due to preferential partitioning to LDs [19,26,27] or through effects on proteasome function [28].

The mechanism by which resident kidney cells accumulate FAs and cholesterol is multifactorial, and include increased lipogenesis [29], increased uptake from extracellular compartments [30,31], and decreased efflux from intracellular to extracellular compartments [32]. β-oxidation may also be decreased [33], though this is an inconsistent finding [34]. Cell-specific mechanisms are discussed in subsequent sections devoted to podocytes and proximal tubule epithelial cells, the two most important cell types in DKD pathogenesis.

There are multiple intracellular pathways that can be activated to minimize lipotoxicity, most notably the sequestration of FAs and cholesterol in cytosolic LDs, which is discussed in detail in a separate section of this review. Unlike adipocytes, where the cytoplasm is almost entirely filled with LDs, epithelial cells have a limited capacity to buffer excess intracellular lipids through LD accrual. Therefore, once the threshold for FA metabolism and storage has been exceeded, the residual FAs exert lipotoxic effects. If these cytoprotective pathways are surmounted, the released lipid metabolites, including ceramides, sphingolipids, diacylglycerols and long-chain acyl-CoAs, can lead to membrane disruption, aberrant lipid modification and malfunction of proteins, such as transporters and signaling enzymes, ROS generation, ER stress, impaired autophagy and increased apoptosis. These downstream sequelae have recently been extensively reviewed [35,36]; this review instead focuses on upstream lipotoxicity triggers, which may be amenable to interventions that abrogate DKD progression.

## 2. Lipotoxicity of Podocytes

Podocytes are highly specialized and terminally differentiated (incapable of self-regeneration) cells, which are critical for normal glomerular function. The loss of podocytes leads to reduced glomerular basement membrane coverage, which manifests initially as albuminuria, and ultimately glomerulosclerosis. In DKD and non-DKD models the accumulation of both cholesterol ester and FA metabolites in podocytes has been implicated in the pathogenesis of glomerular dysfunction (Figure 1).

### 2.1. Podocyte Lipotoxicity Due to Altered Cholesterol Metabolism

Podocytes are unique by virtue of plasma membrane projections that form the slit diaphragm. These membrane domains are enriched with cholesterol-concentrated lipid rafts, which aggregate proteins that regulate the glomerular filtration barrier, such as nephrin, podocin, CD2-associated protein, α-actinin-4 and TRPC6 [37,38]. Multiple studies have demonstrated alterations in plasma membrane lipid components in DKD. Microarray studies from human biopsies identified significant upregulation of mRNA expression of acid sphingomyelinase-like phosphodiesterase 3b (SMPDL3b), which localizes to lipid rafts, and catalyzes sphingomyelin catabolism [39]. Biochemical ramifications included decreased ceramide-1-phosphate and perhaps increased free ceramide [40], which induces apoptosis (Figure 1). Genetic deletion or chemical inhibition of SMPDL3b inhibited podocyte apoptosis and albuminuria in *Lepr^db/db^* mice [39], which phenocopy type 2 diabetes and some aspects of DKD. A subsequent report linked increased SMPDL3b expression to altered caveolin-dependent insulin receptor function [40]. Podocyte-specific deletion of SMPDL3b or supplementation of the downstream product C16:0 ceramide-1-phosphate also rescued podocyte insulin signaling, apoptosis and albuminuria [40]. Interestingly, opposite effects were observed with adriamycin-induced podocyte injury, suggesting a SMPDL3b mechanism may not be common to all podocytopathies, perhaps due to differences in lipid concentrations and exposure.

Two groups have shown that cholesterol accumulates in the cytoplasm of podocytes in humans and animal models of DKD [41,42,43] (Figure 1). Gene expression and functional experiments support both enhanced cholesterol uptake, as well as decreased efflux from decreased ATP-binding cassette A1 (ABCA1) cholesterol transporter activity [41,42,43]. The mechanism by which cholesterol-containing lipoproteins would achieve access to podocyte LDL receptors in vivo remains unclear. A series of studies demonstrated decreased podocyte expression of ABCA1 in mouse (BTBR^ob/ob^, BKS^db/db^ and streptozotocin-treated) models of diabetes [41,42,43]. Increased podocyte ABCA1 expression in transgenic or pharmacologically stimulated mice diminished albuminuria [42,43]. Furthermore, induction of cholesterol efflux with cyclodextrin improved albuminuria, as well as glycemic control, while treatment with HMG-CoA reductase inhibitors had no effect [32], suggesting that decreased cholesterol efflux, rather than enhanced cholesterol synthesis, was the mechanism of cholesterol accumulation. However, recent studies indicate that the ABCA1-dependent mechanism may be due to accumulation of cardiolipin, rather than cholesterol [43]. While these studies collectively support ABCA1-associated podocyte lipotoxicity in DKD pathophysiology, confirmation in DKD animal models that exhibit GFR decline or tubulointerstitial disease [14,44,45], which is tightly associated with GFR decline, is warranted.

### 2.2. Podocyte Lipotoxicity Due to Altered Fatty Acid Metabolism

Podocyte FAs may accumulate due to enhanced uptake, increased synthesis, or decreased degradation [46] (Figure 1). In studies from Fu et al., expression of junctional adhesion molecule-like protein (JAML) was noted to be increased in glomeruli from diabetic patients and two diabetic mouse models, and inversely correlated with GFR [47]. Importantly, podocyte-specific JAML deletion ameliorated lipid accumulation, glomerular histopathology and albuminuria by inhibiting sirtuin-1-dependent, sterol regulatory element-binding protein (SREBP)1-mediated FA synthesis [47].

Exposure of podocytes to the saturated, long-chain saturated FA palmitate is toxic, though compensatory upregulation of desaturase enzymes may initially limit toxicity in DKD [48]. Chung et al. [30] demonstrated that medium and long-chain FA uptake via the G-protein-coupled FFA1 receptor mediated cytoskeletal rearrangements that mimic podocyte foot process effacement (Figure 1). FFA1-dependent FA uptake may also precipitate albuminuria through an indirect mechanism, by inducing angiopoietin-like 4 [49]. FA uptake in podocytes is additionally mediated by the CD36 scavenger receptor; palmitate exposure to cultured podocytes stimulated CD36 expression, as well as ROS-dependent apoptosis [50]. Podocyte FA accumulation was also noted in diabetic BKS^db/db^ mice, due to a mechanism involving paracrine VEGF-B-mediated increase in podocyte FATP4 expression [51].

### 2.3. Lipotoxicity Involving the Tubulointerstitium

Proximal tubule lipid accumulation in DKD was first described by Kimmelstiel [7], and a link between tubule lipotoxicity and DKD is well established in humans [41], as well as rat [52] and mouse [29,33,53,54,55,56] models of type 1 and 2 diabetes. The prevailing model of proximal tubule lipotoxicity is predicated upon leakage of albumin-bound FAs across the injured podocyte and glomerular basement membrane, with subsequent uptake by apical proximal tubule transporters (Figure 2).

Apical proximal tubule glucose reabsorption is mediated by SGLT and GLUT transporters, but the purpose is for glucose reclamation, rather than metabolism. Although capable of modulating to glucose utilization, the proximal tubule is usually gluconeogenic. However, proximal tubules have the capacity to utilize glucose and gluconeogenic substrates for ATP generation [57,58,59,60,61,62,63], and one purported mechanism of SGLT2 inhibitor efficacy is through inhibition of the proximal tubule shift to glucose utilization and glycolysis [61,64]. Under normal circumstances metabolically demanding proximal tubules rely upon FAs, which are more efficient than glucose [65], as the main substrate for ATP generation [66,67]. Circulating FAs are transported across the basolateral membrane by undefined mechanisms (Figure 2). Following uptake, FAs are rapidly converted to acyl-CoA molecules by acyl-CoA synthetases, then preferentially shuttled to the outer mitochondrial membrane, and catalyzed by carnitine palmitoyl transferase-1 (CPT1), the rate-limiting step in FA metabolism. This conversion to acylcarnitine permits FA to traverse the mitochondrial matrix and inner mitochondrial membrane. Conversion of acylcarnitine back to long-chain acyl-CoA by CPT2, and then long-chain FAs, permits FAs to then undergo β-oxidation, ultimately yielding acetyl CoA plus FADH2 and NADH. This is then followed by the synthesis of citrate from acetyl CoA plus oxaloacetate, which can then enter the Krebs cycle, and leads to ATP generation.

Once proximal tubule cell metabolic needs are met, unless the FAs are sequestered, or metabolized without significant ROS generation, excess FAs and toxic metabolites lead to tubular atrophy through mitochondrial and ER stress and lipoapoptosis. The mechanism by which FAs accumulate in the proximal tubule is multifactorial, with evidence for increased uptake [31,41,63,68,69] and increased synthesis, by SREBPs, fatty acid synthase, carbohydrate response element-binding protein (ChREBP), peroxisome proliferator-activated receptor-α (PPARα), and farnesoid X receptor (FXR) deficiency [29,52,53,54,55,56]. These pathways are not mutually exclusive, but likely additive, and the consequence is the accumulation of toxic long-chain FA and metabolites, such as long-chain acyl-CoAs (LC-CoA), ceramides and diacylglycerol (DAG) [13,63,70] (Figure 2). Some [29,33,41], but not all [34,63,71] studies have also shown decreased β-oxidation, which is reflected in metabolomic data from diabetic sera [18].

FAs circulate either esterified to a glycerol backbone as triacylglycerol (TAG), or in a non-esterified form bound to albumin. Under normal circumstances TAG and non-esterified FA are too large to cross the glomerular filtration barrier. However, in “nephrotic” glomerular diseases, of which DKD is a prototype, FA-bound albumin traverses the injured GBM. Therefore, an important consequence of DKD is the aberrant filtration and exposure of albumin-bound long-chain FAs to the apical surface of the proximal tubule (Figure 2). Dextran bead clearance experiments demonstrate that molecules with an effective radius equivalent to albumin (3.5 nm) are permeable to glomeruli with DKD [72,73], whereas TAG within lipoproteins (>11 nm) would still be too large. Although DKD may progress without albuminuria [74], it remains one of the biggest risks for progression to end stage kidney disease [75]. Albumin per se is not cytotoxic to the proximal tubule [13,68,69,76], but FA bound to albumin instigate apoptosis in the proximal tubule [9,10,11] (Figure 2), as well as in other tissues [3,10,34,77].

Approximately 50% of apical proximal tubule NEFA uptake is mediated by the high affinity, low capacity transporter, FATP2 [31]. Under normal circumstances FATP2 is a scavenger for low concentrations of filtered FAs. However, in pathologic states, in which the glomerular filtration barrier is injured, permitting albumin-bound FA filtration and exposure to the AP proximal tubule membrane, consequent FATP2-dependent FA uptake leads to proximal tubular epithelial cell apoptosis and tubular atrophy [31]. As previously mentioned, proximal tubules utilize FAs as the major substrate for ATP generation, and multiple groups have determined that β-oxidation is diminished in DKD [29,33,41], resulting in decreased ATP generation in this energy-demanding nephron segment. Furthermore, when unmetabolized FAs are combined with enhanced FATP2-dependent FA uptake, intracellular FAs and FA metabolites, such as long-chain acyl-CoAs, accumulate and contribute to lipotoxicity [13]. 

Global FATP2 gene (*Slc27a2*) deletion combined with genetic or inducible mouse models that phenocopy progressive DKD, resulted in abrogation of tubular atrophy and restoration of GFR [14]. Although amelioration of DKD could be due to deletion of proximal tubule FATP2, the global FATP2 knockout mice also achieved dramatic reductions in plasma glucose, which may exert indirect kidney benefits [78]. Furthermore, FATP2 deletion does not completely eliminate apical proximal tubule FA uptake [31]. The residual FA transporters are incompletely characterized; FATP3 is also expressed along the apical proximal tubule membrane, though not nearly as abundantly as FATP2 (Khan et al., unpublished observations). In addition to mediating bulk FA uptake, FATP2 physically interacts with ceramide synthase 2, and contributes catalytically to increased synthesis of potentially cytotoxic ceramides and dihydroceramides [79].

FATP2 has also been implicated in the pathogenesis of non-diabetic kidney diseases. In a mouse ureteral obstruction model, FATP2-dependent FA uptake was enhanced, leading to cytokine release and fibrogenesis [80]. In mouse models of zoledronate-induced renal toxicity, and in vitro cell culture confirmation, increased proximal tubule saturated FA accumulation and fibrosis were mediated by a TGFβ-dependent increase in FATP2 expression [81]. In a study of renal transplant biopsies, enhanced FATP2 mRNA expression was associated with calcineurin inhibitor-induced interstitial fibrosis and tubular atrophy [82].

Among FA transporters not in the FATP family, Kidney Injury Molecule-1 (KIM-1), also known as T cell immunoglobulin mucin domain (TIM)-1, has recently been implicated in the pathogenesis of proximal tubule lipotoxicity. KIM-1 is expressed within the apical proximal tubule membrane and serves as a scavenger receptor for oxidized lipoproteins, as well as for apoptotic cells, through recognition of externalized phosphatidylserine [83] (Figure 2). Apical proximal tubule KIM-1 expression is increased with tubular injury [84], and constitutive overexpression caused interstitial fibrosis and GFR decline [85]. Clinically, plasma KIM-1 is a reliable biomarker for DKD [86,87,88]. In vitro and DKD mouse model studies revealed that KIM-1 mediated uptake of albumin-bound palmitate by proximal tubules [89]. Whether KIM-1 directly or indirectly transports FA was not established [89]. KIM-1-regulated FA uptake in a DKD model caused significant albuminuria, tubular atrophy, interstitial fibrosis, glomerulosclerosis, as well as secretion of inflammatory and fibrosis-inducing cytokines [89]. KIM-1-related effects on GFR were not evaluated. Consistent results were observed in the obese Zucker rat model of DKD, where cDNA microarray revealed that KIM-1 was the most highly upregulated transcript [90].

Endocannabinoids are small lipids derived from plasma membrane phospholipids, which bind in a paracrine fashion to G protein-coupled CB1 and CB2 receptors, and mediate opposing effects. Although both receptor types, as well as ligand-degrading enzymes are expressed throughout the nephron, CB1 predominates in the proximal tubule and podocytes [91,92]. In mouse and rat models of DKD, CB1 expression is upregulated in podocytes, and associated with albuminuria [93,94]. In the proximal tubule, palmitate induced proximal tubule CB1 expression, and potentiated apoptosis, suggesting that CB1 could mediate proximal tubule lipotoxicity [95]. In mice on a high fat diet, proximal tubule-specific deletion of CB1 ameliorated many aspects of DKD, including albuminuria, modest changes in GFR and interstitial fibrosis [96]. One purported mechanism was through enhanced AMP-activated protein kinase (AMPK)-mediated autophagy and enhanced FA β-oxidation [96]. These studies have led to multiple pre-clinical trials with CB1 inhibitors for DKD (see Table 1 and associated text). Studies with CB2 are more limited, though global CB2 deletion worsened albuminuria, serum creatinine and glomerulosclerosis in STZ-treated mice [97]. Although CB2 expression has been noted in cultured cells and rodent kidney [98], the lack of CB2 detection in human kidney [91] raises the issue whether the global CB2 knockout renal phenotype could be derived from extrarenal effects.

In microarray experiments from human diabetic kidneys, proximal tubule CD36 was highly expressed, stimulated by glucose, and regulated palmitate-induced apoptosis [99]. However, kidney CD36 mRNA expression was decreased or absent in mouse models of DKD [89,100,101]. To rectify these discrepancies kidney-specific CD36 transgenic mice were induced to develop diabetes with streptozotocin [33]. Although renal function was not assessed, interstitial fibrosis was not observed in the diabetic, CD36 transgenic mice. The absence of pathology was despite proximal tubule cytosolic LD accumulation, suggesting that CD36 may regulate FA compartmentalization to LDs, thereby shielding cells from lipotoxicity. Taken together, the data indicate that CD36 does not mediate proximal tubule lipotoxicity or tubular atrophy in DKD.

### 2.4. Role of Lipid Droplets

Numerous pathways buffer against FA accumulation and subsequent lipotoxicity. The initial response to FA uptake is metabolism by β-oxidation to generate ATP [66]. An additional defense against lipotoxicity is the Them/ACOT family of thioesterases, which accelerate catabolism of LC-CoAs, the initial FA metabolite, by hydrolyzing the thioester bond to form CoA and the corresponding FA [102,103]. Once energy needs are met, excess FAs and cholesterol are compartmentalized as TAG and cholesterol ester aggregates within the LD core [104], which is surrounded by a single phospholipid membrane. The LD membrane is decorated with proteins that facilitate LD formation, such as diacylglycerol transferase-2 (DGAT2) [105], and maintenance, such as perilipins [106], which block TAG release by inhibiting adipose triglyceride lipase (ATGL).

While all cells are capable of incorporating TAG into LDs [107], which primarily function as reservoirs for energy demands by facilitating bidirectional FA shuttling, LDs are not prominent in the cytosol of epithelial cells under normal circumstances. It is therefore reasonable to equate easily detectable LDs with a pathologic state. However, it is important to note that LD expansion represents a compensatory, protective pathway in most organs, including kidney and kidney model systems [108,109,110].

The simplest view is that LDs serve as depots for excess lipids, and thereby shield cells from “free” cholesterol and FA metabolites, by preventing lipotoxicity and lipoapoptosis [108,111,112] or by enhancing autophagy pathways [113]. A more nuanced view is predicated upon LD proximity to other organelles, which are dynamically motile within the cytosol [114]. LDs tethered to elongated mitochondria that have undergone fusion, results in enhanced generation of ATP, which fuels lipogenesis and LD expansion [115,116]. This process involves metabolic reprogramming to utilize glucose/pyruvate as a substrate, although a recent report suggests that the mitochondrial-LD juxtaposition merely enhances FA β-oxidation efficiency [117].

Two recent reports from the Pollak group revealed that FA-induced LD expansion in podocytes led to preferential translocation of APOL1 G1 and G2 risk variants from ER to LDs, which was associated with reduced cytotoxicity [118,119]. Although these studies illuminate an intriguing cytoprotective role of LDs in podocytes, it should be noted that APOL1 risk variants have been associated primarily with non-DKD [120].

Because kidney LDs have been associated with DKD (and other proteinuric renal diseases), a common strategy has been to reduce LD mass through enhanced FA β-oxidation. However, this approach risks forced mitochondrial entry of FA, which could lead to oxidative stress. Alternatively, enhanced FA sequestration into LDs could be beneficial, but unlike adipocytes, which have tremendous capacity to accommodate LDs in the cytosol, epithelial cells have limits.

## 3. Potential Lipotoxicity Treatments

### 3.1. Drugs in Current Clinical Use

#### 3.1.1. Fibrates

**Table 1 cells-11-03236-t001:** Candidate drugs for the treatment of renal lipotoxicity.

Drug	Target	Mechanism	References
Fenofibrate	PPARα	FA uptake and metabolism	[121,122,123,124,125,126]
Statins	HMG-CoA reductase	Cholesterol synthesis	[127,128,129,130,131]
SGLT2 inhibitors	SGLT2	Unknown	[132,133,134,135,136,137,138,139,140]
Liraglutide	GLP1 receptors	Unknown	[141]
A30	ABCA1	Cholesterol efflux	[43]
Lipofermata	FATP2	FA uptake	[80]
PBI-4050	GPR40/GPR84	FA uptake	[142,143]
TW-37	KIM-1	FA uptake	[89]
AM-251, rimonabant	CB1	FA metabolism	[93,144,145,146]
INT-767	FXR/TGR5	FA synthesis	[147]
AdipoRon	AdipoR1/R2	FA metabolism	[148]
Dehydrozingerone	Multiple	FA synthesis	[149]
Resveratrol	SIRT1, AMPK	FA synthesis and metabolism	[150,151]

Several studies, in both animal models and humans, have examined the effect of fenofibrate on DKD progression. The rationale is that fibrates are activators of PPARα, which regulates FA catabolism at several levels, and theoretically reduces the propensity for lipotoxicity. In vitro studies support that decreased PPARα expression is associated with lipoapoptosis and fibrogenesis [152]. In experiments with C57BLKS *Lepr^db/db^* mice fenofibrate improved glucose tolerance and oxidative stress, as well as albuminuria and glomerular pathology [153,154]. In a non-DKD (folic acid nephropathy) model, fenofibrate normalized serum creatinine [33]. The fenofibrate effects may be partly mediated through AMPK activation and enhanced autophagy [155].

However, interpretation of fenofibrate human clinical trials is more complicated, and results are not as positive as with animal studies. In two large diabetes trials, FIELD and ACCORD, randomization to fenofibrate improved albuminuria, but caused an initial small increase in creatinine, which persisted throughout the study [121,122], and was reversible upon fenofibrate discontinuation [123]. A follow-up trial in a FIELD subcohort showed a fenofibrate-associated increase in both serum creatinine and cystatin C (indicating that the creatinine changes are not artifactual, and represent true changes in GFR), as well as no benefit in albuminuria reduction [124]. The mechanism for the initial creatinine increase is unclear, but may be related to fenofibrate-induced inhibition of vasodilator prostaglandins [125]. Despite the initial effects on GFR, the rate of eGFR decline over the course of the ACCORD trial was significantly lower in the group randomized to fenofibrate [126]. Newer PPARα activators, which are not metabolized by kidney, may represent safer alternatives to treat renal lipotoxicity [156]. 

One explanation for the discrepancy of the animal and human results is that PPARα affects the transcription of multiple lipid metabolism genes, both protective and deleterious (e.g., FATP2). In addition, effects on GFR are strain-dependent in mice, and may not be manifest in mouse DKD models with forgiving genetic backgrounds.

#### 3.1.2. HMG-CoA Reductase Inhibitors (Statins)

Statins are among the most widely prescribed drugs in the world, with demonstrated cardiovascular benefits in patients with CKD [127], though the effects are attenuated with lower baseline eGFRs [128]. The effect of statins specifically on DKD is less clear. Two recent meta-analyses of randomized controlled trials showed that statins were associated with modest reduction of albuminuria, but not with modulation of eGFR decline [129,130]. These data are consistent with a larger meta-analysis of DKD and non-DKD patients, which noted no effect on eGFR in the DKD subgroup analysis [131].

#### 3.1.3. Proprotein Convertase Subtilisin-Kexin Type 9 (PCSK9) Inhibitors

Inhibitors of PCSK9 (evolocumab and alirocumab) are now recognized as alternatives or adjuncts to statins for LDL-cholesterol reduction. The PCSK9 gene is expressed at very low levels in the kidney, with modest modulation in the loop of Henle and collecting duct with diabetes [101]. A recent review concludes that PCSK9 inhibitors are safe and effective for cholesterol reduction with modest CKD (including DKD), though similar data are unavailable for patients with eGFR < 30 mL/min/1.73 m^2^ [157]. There are also insufficient data regarding the role of PCSK9 inhibitors in renal lipotoxicity or DKD progression.

#### 3.1.4. Sodium-Glucose Transporter-2 Inhibitors (SGLT2i)

SGLT2i are now recognized as the standard of care for patients with DKD and eGFR >25 mL/min/1.73 m^2^, and adjunctive to glycemic control and renin-angiotensin-aldosterone antagonist therapies [132,133,134]. However, regiments that include SGLT2i-induced prevention of DKD progression is incompletely established, and likely to be multi-factorial, including some evidence for anti-lipotoxic effects [135]. Dapagliflozin prevented renal tubule lipid accumulation and fibrosis in mice maintained on a high fat diet [136]. In the FLS-*ob*/*ob* mouse model of diabetes, ipragliflozin also reduced tubular lipid deposition, and was associated with decreased apoptosis and ER stress [137]. Comparable results were observed with dapagliflozin in *Lepr^db/db^* mice [138]. Two additional studies in high fat diet-fed mice demonstrated that SGLT2i preserved renal tubule mitochondrial morphology and function [139,140].

#### 3.1.5. Glucagon-like Peptide-1 (GLP-1) Receptor Agonists

Like SGLT2 inhibitors liraglutide has been shown to have renoprotective effects in clinical trials, but the benefit is mainly limited to reduction in proteinuria. In preclinical studies, a 12-week course of liraglutide improved plasma lipid profiles and reduced proximal tubule lipid droplets in an inducible rat model of DKD [141], suggesting that it might prevent lipotoxicity and renal dysfunction.

#### 3.1.6. Mineralocorticoid Receptor Antagonists (MRA)

Inhibition of the renin-angiotensin-aldosterone cascade has been a standard treatment for DKD for decades. While the pathophysiologic focus has largely been on angiotensin II, aldosterone has independent effects, which include inflammation and fibrosis [158]. Recent studies have shown that the non-steroidal MRA, finerenone, may be benefical when added to renin-angiotensin inhibitors, for slowing the progression of DKD [159,160,161]. There are multiple reports of spironolactone and eplerenone efficacy for hepatosteatosis and liver fibrosis in mouse models [162,163,164], but data are lacking regarding the impact of MRA on renal lipotoxicity.

### 3.2. Experimental Compounds

#### 3.2.1. ABCA1 Induction

In conjunction with experiments to characterize the role of ABCA1 in podocyte protection, Ducasa et al. employed a previously described inducer of ABCA1 expression (A30) in their in vivo studies [43]. *Lepr^db/db^* mice treated with A30 for two to four weeks demonstrated restoration of ABCA1 expression, as well as reductions in albuminuria and BUN. Histologic analyses revealed decreased cortical cholesterol, FA and peroxidized cardiolipin accumulation, and improved glomerular histopathology [43]. Other small molecule (pyridine carboxamide) inducers of ABCA1 are under investigation as potential therapies for DKD, as well as for FSGS and Alport’s syndrome.

Sterol-O-acyltransferase-1 (SOAT1) catalyzes the esterification of cholesterol to cholesterol esters, which are then stored in LDs. Pharmacologic SOAT1 inhibition with Sandoz 58-035 improved albuminuria, serum creatinine and podocyte LD accumulation in a mouse model of Alport’s syndrome [165]. In vitro studies in culture podocytes revealed that SOAT1 inhibition was cytoprotective, by a mechanism that involved enhanced ABCA1 expression [165].

#### 3.2.2. FATP2 Inhibitors

Two small molecular weight FATP2 inhibitors (lipofermata and grassofermata) were identified from high throughput screens, with low μM IC_50_ in fluorescent FA uptake assays [166]. Although pharmacokinetic data are limited, in a mouse model of urinary tract obstruction, oral administration of lipofermata prevented tubule lipid accumulation and fibrosis; renal function studies were not conducted [80].

#### 3.2.3. G-protein-Coupled FA Receptor Inhibitors

GP40 (also known as FFAR1) and GP84 are G-protein-couple FA receptors expressed predominantly in podocytes in kidney [14], though mRNA is detectable in proximal tubule and collecting duct [142]. PBI-4050 is a synthetic medium-chain fatty acid analog with agonist and antagonist activity toward GPR40 and GPR84, respectively. In experiments with the *Lepr^db/db^ eNOS*^-/-^ model of type 2 diabetes and progressive kidney disease, PBI-4050 was effective in primary prevention and as treatment for established DKD, as determined by reductions in albuminuria, GFR decline and glomerulosclerosis [142]. Interestingly, PBI-4050 also improved glucose tolerance by preserving pancreatic β-cell function and insulin secretion [142], similar to the phenotype observed in FATP2 knockout mice [14]. Additional studies in GPR40 and GPR84 knockout mice crossed with non-diabetic models of interstitial fibrosis revealed that GPR40 was protective, whereas GPR84 was deleterious. PBI-4050 also inhibited fibrosis in these non-diabetic models [143].

GPR43 (also known as FFAR2) is a short-chain FA receptor. In mice with type 1 diabetes induced by streptozotocin, superimposed GPR43 deletion attenuated LDL receptor-mediated cholesterol accumulation in podocytes [167]. Further studies in cultured podocyte identified autophagy inhibition and ERK-dependent activation of the EGR1 transcription factor as potential mechanisms. Treatment of cultured podocytes with the ERK 1/2 inhibitor PD0325901 decreased LDL receptor expression and cholesterol accumulation, and enhanced autophagy [167].

#### 3.2.4. KIM-1 Inhibition

Mori et al. conducted a high throughput screen of a 14,430 compound small molecule library, using inhibition of oxidized LDL uptake as the readout in high KIM-1 expressing kidney-derived 769-P cells [89]. These experiments yielded TW-37 as a candidate KIM-1 inhibitor. Extensive in vitro and in vivo (non-DKD, palmitate complexed to albumin overload model) studies revealed that TW-37 inhibited KIM-1-mediated FA uptake, and blunted interstitial inflammation and fibrosis [89].

#### 3.2.5. CB1 Inhibition

AM-251 and rimonabant are structurally similar biarylpyrazole compounds with inverse agonist/antagonist activity for the CB1 cannabinoid receptor. In animal models, rimonabant reduced albuminuria and glomerulosclerosis in obese fa/fa Zucker rats [144], and decreased albuminuria and kidney lipid content in *Lepr^db/db^* mice [145]. AM-251 also reduced albuminuria in streptozotocin-treated mice and a rat model of obesity [93,146]. Jourdan et al. demonstrated improved albuminuria and GFR in ZDF rats with another CB1 inverse agonist, JD5037 [94]. Rimonabant is being tested for effects on glycemia and obesity in several clinical trials, though none are measuring DKD outcomes.

#### 3.2.6. FXR/TGR5 Activation

The FXR nuclear hormone and TGR5 G protein–coupled bile acid receptors have been implicated in the pathogenesis of DKD [55,168]. RNAseq analysis (and targeted protein expression confirmation) of kidneys from streptozotocin-induced diabetic mice revealed that treatment with the selective FXR (INT-747) or TGR5 (INT-777) agonists affect distinct pathways [147]; INT-747 uniquely increased SREBP-1-mediated lipogenesis, whereas INT-777 enhanced mitochondrial biogenesis. Importantly, the combined FXR/TGR5 agonist (INT-767) decreased cortical FA and cholesterol accumulation, and dramatically reduced albuminuria, as well as interstitial and glomerular fibrosis in both the streptozotocin and *Lepr^db/db^* diabetic mouse models [147].

#### 3.2.7. Adiponectin Receptor Activation

Adiponectin is a cytokine that is secreted by adipocytes, and has been associated with amelioration of diabetic effects through anti-inflammatory, anti-fibrotic, and anti-oxidant activities. Adiponectin regulates FA metabolism by inducing AMPK phosphorylation and increasing peroxisome PPARα expression. AdipoR1 and AdipoR2 adiponectin receptors are expressed throughout the human kidney, but glomerular expression was decreased in DKD, coinciding with early GFR decline [148]. AdipoRon is a synthetic adiponectin receptor agonist which binds to both AdipoR1 and AdipoR2 receptors [169]. AdipoRon treatment for four weeks significantly reduced albuminuria in *Lepr^db/db^* mice, and was associated with enhanced Ca^2+^/calmodulin-dependent protein kinase, liver kinase B1 and AMPK enzyme activities, as well as increased PPARα expression [148].

#### 3.2.8. Dehydrozingerone

Dehydrozingerone (DHZ) is a curcumin analog that exhibits anti-neoplastic, anti-oxidant and anti-diabetic effects. C57BL/6 mice on a high fat diet and treated with DHZ for four weeks experienced significant reductions in albuminuria, as well as reduced kidney FA and cholesterol content, and improved glomerular histology [149]. Expression patterns derived from quantitative PCR analysis of kidney samples suggested several mechanisms for DHZ efficacy, including reduced FA synthesis and decreased oxidative stress.

#### 3.2.9. Resveratrol

Resveratrol is a polyphenol compound that is abundant in grapes and red wine, which can also be taken as a dietary supplement. Purported metabolic benefits are myriad, including reduced serum glucose and lipid concentrations. Pre-clinical studies suggest resveratrol regulates multiple mechanisms, including reduction in FA synthesis, inflammation, oxidative and ER stress, mitochondrial dysfunction, autophagy and apoptosis [170]. In a recent study with high fat diet-fed C57BL/6J mice, resveratrol supplementation decreased blood glucose and lipid concentrations, and was associated with a trend toward lower serum creatinine [150]. However, a recent meta-analysis of randomized clinical diabetes trials showed a significant benefit of resveratrol for lower systolic blood pressure, but no effect on serum creatinine, though there was a paucity of studies evaluating relevant DKD biomarkers [151].

## 4. Conclusions

Lipotoxicity is an increasingly recognized mechanism of DKD pathophysiology in the podocyte and proximal tubule. Because multiple pathways have been identified, and the mechanisms of podocyte and proximal tubule lipotoxicity may differ (Figure 1 and Figure 2), there may be opportunities to cultivate multiple kidney therapeutic targets, e.g., different glomerular and tubular therapies [171]. This strategy could conceivably result in therapeutic synergism, much like the combined effects of ACE inhibitor or ARBs with SGLT2 inhibitors for DKD, or chemotherapy protocols.

## Figures and Tables

**Figure 1 cells-11-03236-f001:**
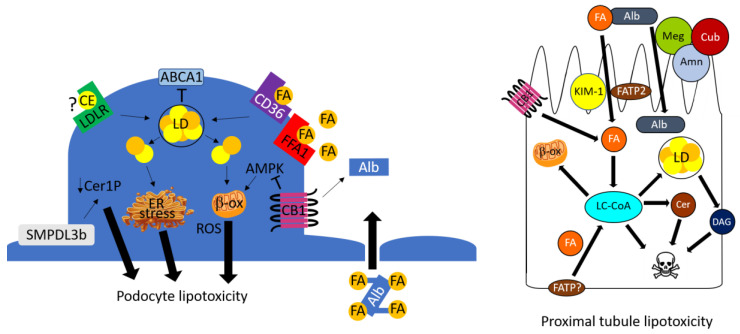
Lipotoxicity in the podocyte. Under pathologic (albuminuric) circumstances, fatty acids and cholesterol esters are taken up by separate mechanisms. If the accumulated lipids exceed storage capacity within lipid droplets, toxic metabolites accumulate and lead to reactive oxygen species generation, endoplasmic reticulum stress, and ultimately podocyte cell death. Abbreviations: albumin (Alb), AMP-activated protein kinase (AMPK), ATP-binding cassette A1 (ABCA1), cannabinoid receptor type 1 (CB1), ceramide-1-phosphate (Cer1P), cholesterol ester (CE), endoplasmic reticulum (ER), fatty acid (FA), free fatty acid receptor-1 (FFA1), lipid droplet (LD), low density lipoprotein receptor (LDLR), reactive oxygen species (ROS), acid sphingomyelinase-like phosphodiesterase 3b (SMPDL3b).

**Figure 2 cells-11-03236-f002:**
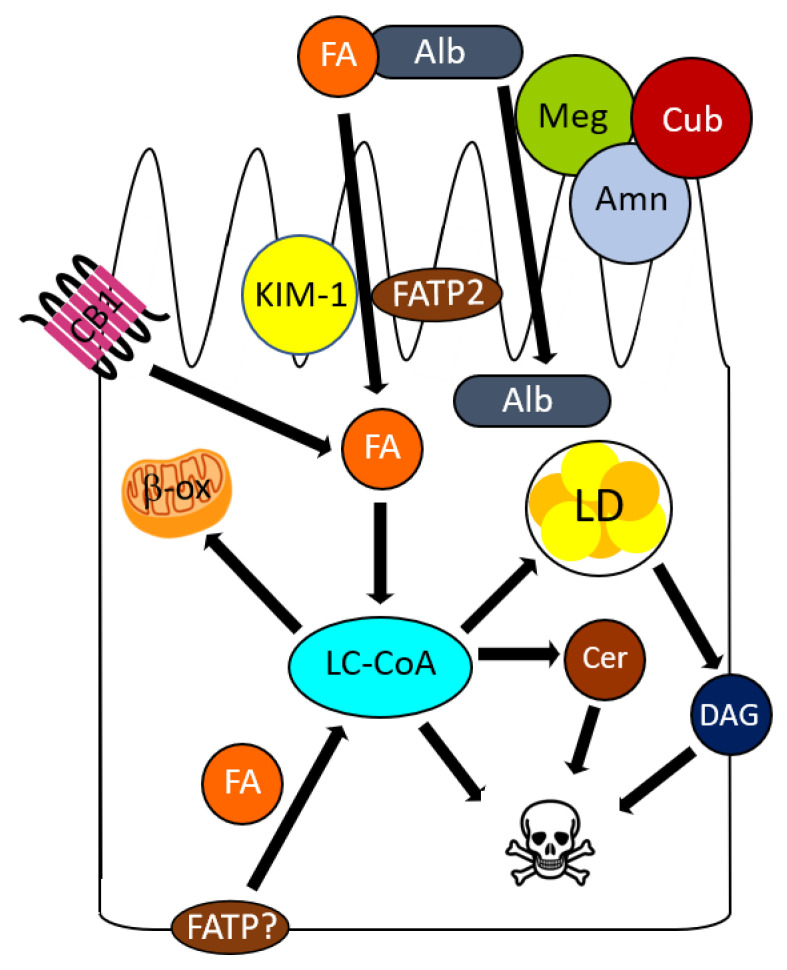
Lipotoxicity in the proximal tubule cell. Under pathologic (albuminuric) circumustances, fatty acids are taken up by transporters at the basolateral and apical membranes. At the apical surface, fatty acids and albumin dissociate and are transported by separate mechanisms. If the accumulated lipids exceed the capacity to undergo β-oxidation within mitochondria or storage within lipid droplets, toxic fatty acid metabolites (ceramides, diacylglycerols, long-chain acyl-CoAs) cause apoptosis and tubular atrophy. Abbreviations: albumin (Alb), amnionless (Amn), cannabinoid receptor type 1 (CB1), ceramide (Cer), cubilin (Cub), diacylglycerols (DAG), fatty acid (FA), fatty acid transport protein (FATP), kidney injury molecule-1 (KIM-1), lipid droplet (LD), low density lipoprotein receptor (LDLR), long-chain acyl-CoAs (LC-CoA), megalin (Meg).

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
