# Peer review of "The Contribution of Lipotoxicity to Diabetic Kidney Disease"

_cells, 2022, doi:10.3390/cells11203236_

Round 1

Reviewer 1 Report

Schelling has submitted a comprehensive review of the field of kidney lipotoxicity.  This is an outstanding manuscript covering considerable material concisely yet comprehensively.

This reviewer has only 3 considerations for the author

1.  The table showing potential therapies for lipotoxicity as well as targets is very informative.  The author could consider adding which mechanism is targeted, e.g., uptake of fatty acids, etc.

2.  The author touched on cholesterol only minimally.  PCSK9 is expressed in kidney.  What does the author think that this protein may be doing with regard to lipid metabolism in the proximal tubule?

3.  As a related note, the author did not touch on StAR related proteins which also play a role in lipid and cholesterol trafficking.  What role do these proteins play in lipotoxicity?

Author Response

Reviewer 1 Schelling has submitted a comprehensive review of the field of kidney lipotoxicity. This is an outstanding manuscript covering considerable material concisely yet comprehensively. This reviewer has only 3 considerations for the author 1. The table showing potential therapies for lipotoxicity as well as targets is very informative. The author could consider adding which mechanism is targeted, e.g., uptake of fatty acids, etc. Thanks for the suggestion. An additional column was added to the table, which depicts purported mechanisms. 2. The author touched on cholesterol only minimally. PCSK9 is expressed in kidney. What does the author think that this protein may be doing with regard to lipid metabolism in the proximal tubule? A short paragraph devoted to PCSK9 inhibitors has been added (p.15, line 18 – p.16, line 2). 3. As a related note, the author did not touch on StAR related proteins which also play a role in lipid and cholesterol trafficking. What role do these proteins play in lipotoxicity Although StAR proteins affect cholesterol transport across mitochondrial membranes, the main effect of StAR perturbations is on steroid hormone synthesis and secretion. Up- or down-regulation of steroid hormone levels could affect kidney function, but it seems a bit tangential to the lipotoxicity topic. As a result, we respectively chose to not include a section about StAR-related proteins.

Reviewer 2 Report

Comments to the Authors

The Author thoroughly reviewed the lipotoxicity triggers in podocyte and proximal tubules, the current clinical drug use and potential experimental compounds.

Overall, this review is well designed and written.

I recommend acceptance of this review after addressing the comments listed below:

·         Both figures should be revised and prepared more properly! (For example- LDs and mitochondria should appear the same in both figures, the apical membrane of the proximal tubular cell should not be dashed, the name of the cells should appear in the figures, etc.)   

·         In page 7, regarding the mechanisms of lipid accumulation in the proximal tubules- the involvement of downregulated AMPK phosphorylation, consequently reducing β-oxidation and enhanced lipogenesis [1-5], should be added.

·         Although hinted along the review, it is worthwhile mentioning the metabolic shift proximal tubules undergo from utilizing fatty acid oxidation as their main energy source, toward glycolysis [6-8], under diabetic conditions.

·         In the section of the experimental compounds, I would suggest adding the information about cannabinoid receptor 1 (CB1R) antagonists. It is well established that the endocannabinoid/ CB1R system is involved in DKD progression and that genetic deletion or CB1R blocked can ameliorate lipotoxicity in both proximal tubules [5,9,10] and podocyte [11,12].

1.            Han, Y.; Xiong, S.; Zhao, H.; Yang, S.; Yang, M.; Zhu, X.; Jiang, N.; Xiong, X.; Gao, P.; Wei, L.; et al. Lipophagy deficiency exacerbates ectopic lipid accumulation and tubular cells injury in diabetic nephropathy. Cell death & disease 2021, 12, 1031, doi:10.1038/s41419-021-04326-y.

2.            Jeong, H.Y.; Kang, J.M.; Jun, H.H.; Kim, D.J.; Park, S.H.; Sung, M.J.; Heo, J.H.; Yang, D.H.; Lee, S.H.; Lee, S.Y. Chloroquine and amodiaquine enhance AMPK phosphorylation and improve mitochondrial fragmentation in diabetic tubulopathy. Scientific reports 2018, 8, 8774, doi:10.1038/s41598-018-26858-8.

3.            Muratsubaki, S.; Kuno, A.; Tanno, M.; Miki, T.; Yano, T.; Sugawara, H.; Shibata, S.; Abe, K.; Ishikawa, S.; Ohno, K.; et al. Suppressed autophagic response underlies augmentation of renal ischemia/reperfusion injury by type 2 diabetes. Scientific reports 2017, 7, 5311, doi:10.1038/s41598-017-05667-5.

4.            Sohn, M.; Kim, K.; Uddin, M.J.; Lee, G.; Hwang, I.; Kang, H.; Kim, H.; Lee, J.H.; Ha, H. Delayed treatment with fenofibrate protects against high-fat diet-induced kidney injury in mice: the possible role of AMPK autophagy. American journal of physiology. Renal physiology 2017, 312, F323-F334, doi:10.1152/ajprenal.00596.2015.

5.            Udi, S.; Hinden, L.; Earley, B.; Drori, A.; Reuveni, N.; Hadar, R.; Cinar, R.; Nemirovski, A.; Tam, J. Proximal Tubular Cannabinoid-1 Receptor Regulates Obesity-Induced CKD. Journal of the American Society of Nephrology : JASN 2017, 28, 3518-3532, doi:10.1681/ASN.2016101085.

6.            Hinden, L.; Kogot-Levin, A.; Tam, J.; Leibowitz, G. Pathogenesis of diabesity-induced kidney disease: role of kidney nutrient sensing. The FEBS journal 2021, doi:10.1111/febs.15790.

7.            Cai, T.; Ke, Q.; Fang, Y.; Wen, P.; Chen, H.; Yuan, Q.; Luo, J.; Zhang, Y.; Sun, Q.; Lv, Y.; et al. Sodium-glucose cotransporter 2 inhibition suppresses HIF-1alpha-mediated metabolic switch from lipid oxidation to glycolysis in kidney tubule cells of diabetic mice. Cell Death Dis 2020, 11, 390, doi:10.1038/s41419-020-2544-7.

8.            Li, J.; Liu, H.; Takagi, S.; Nitta, K.; Kitada, M.; Srivastava, S.P.; Takagaki, Y.; Kanasaki, K.; Koya, D. Renal protective effects of empagliflozin via inhibition of EMT and aberrant glycolysis in proximal tubules. JCI insight 2020, 5, doi:10.1172/jci.insight.129034.

9.            Lim, J.C.; Lim, S.K.; Han, H.J.; Park, S.H. Cannabinoid receptor 1 mediates palmitic acid-induced apoptosis via endoplasmic reticulum stress in human renal proximal tubular cells. Journal of cellular physiology 2010, 225, 654-663, doi:10.1002/jcp.22255.

10.          Drori, A.; Permyakova, A.; Hadar, R.; Udi, S.; Nemirovski, A.; Tam, J. Cannabinoid-1 receptor regulates mitochondrial dynamics and function in renal proximal tubular cells. Diabetes, obesity & metabolism 2019, 21, 146-159, doi:10.1111/dom.13497.

11.          Nam, D.H.; Lee, M.H.; Kim, J.E.; Song, H.K.; Kang, Y.S.; Lee, J.E.; Kim, H.W.; Cha, J.J.; Hyun, Y.Y.; Kim, S.H.; et al. Blockade of cannabinoid receptor 1 improves insulin resistance, lipid metabolism, and diabetic nephropathy in db/db mice. Endocrinology 2012, 153, 1387-1396, doi:10.1210/en.2011-1423.

12.          Jourdan, T.; Szanda, G.; Rosenberg, A.Z.; Tam, J.; Earley, B.J.; Godlewski, G.; Cinar, R.; Liu, Z.; Liu, J.; Ju, C.; et al. Overactive cannabinoid 1 receptor in podocytes drives type 2 diabetic nephropathy. Proceedings of the National Academy of Sciences of the United States of America 2014, 111, E5420-5428, doi:10.1073/pnas.1419901111.

Author Response

Reviewer 2 
Comments to the Authors 

The Author thoroughly reviewed the lipotoxicity triggers in podocyte and proximal tubules, the current clinical drug use and potential experimental compounds. 
Overall, this review is well designed and written. 
I recommend acceptance of this review after addressing the comments listed below:

· Both figures should be revised and prepared more properly! (For example- LDs and mitochondria should appear the same in both figures, the apical membrane of the proximal tubular cell should not be dashed, the name of the cells should appear in the figures, etc.) 

Both figures were revised, as suggested. 

· In page 7, regarding the mechanisms of lipid accumulation in the proximal tubules- the involvement of downregulated AMPK phosphorylation, consequently reducing β-oxidation and enhanced lipogenesis [1-5], should be added. 

AMPK-mediated mechanisms are now mentioned on p.11, line 18; p.14, lines 9-10; p.19, lines 3 and 9, as well as Figure 1. 

· Although hinted along the review, it is worthwhile mentioning the metabolic shift proximal tubules undergo from utilizing fatty acid oxidation as their main energy source, toward glycolysis [6-8], under diabetic conditions. 

The capacity for proximal tubule shift to glycolysis is now mentioned on p.8, lines 2-9. 

· In the section of the experimental compounds, I would suggest adding the information about cannabinoid receptor 1 (CB1R) antagonists. It is well established that the endocannabinoid/ CB1R system is involved in DKD progression and that genetic deletion or CB1R blocked can ameliorate lipotoxicity in both proximal tubules [5,9,10] and podocyte [11,12]. 

1. Han, Y.; Xiong, S.; Zhao, H.; Yang, S.; Yang, M.; Zhu, X.; Jiang, N.; Xiong, X.; Gao, P.; Wei, L.; et al. Lipophagy deficiency exacerbates ectopic lipid accumulation and tubular cells injury in diabetic nephropathy. Cell death & disease 2021, 12, 1031, doi:10.1038/s41419-021-04326-y. 
2. Jeong, H.Y.; Kang, J.M.; Jun, H.H.; Kim, D.J.; Park, S.H.; Sung, M.J.; Heo, J.H.; Yang, D.H.; Lee, S.H.; Lee, S.Y. Chloroquine and amodiaquine enhance AMPK phosphorylation and improve mitochondrial fragmentation in diabetic tubulopathy. Scientific reports 2018, 8, 8774, doi:10.1038/s41598-018-26858-8. 
3. Muratsubaki, S.; Kuno, A.; Tanno, M.; Miki, T.; Yano, T.; Sugawara, H.; Shibata, S.; Abe, K.; Ishikawa, S.; Ohno, K.; et al. Suppressed autophagic response underlies augmentation of renal ischemia/reperfusion injury by type 2 diabetes. Scientific reports 2017, 7, 5311, doi:10.1038/s41598-017-05667-5. 
4. Sohn, M.; Kim, K.; Uddin, M.J.; Lee, G.; Hwang, I.; Kang, H.; Kim, H.; Lee, J.H.; Ha, H. Delayed treatment with fenofibrate protects against high-fat diet-induced kidney injury in mice: the possible role of AMPK autophagy. American journal of physiology. Renal physiology 2017, 312, F323-F334, doi:10.1152/ajprenal.00596.2015. 
5. Udi, S.; Hinden, L.; Earley, B.; Drori, A.; Reuveni, N.; Hadar, R.; Cinar, R.; Nemirovski, A.; Tam, J. Proximal Tubular Cannabinoid-1 Receptor Regulates Obesity-Induced CKD. Journal of the American Society of Nephrology : JASN 2017, 28, 3518-3532, doi:10.1681/ASN.2016101085. 
6. Hinden, L.; Kogot-Levin, A.; Tam, J.; Leibowitz, G. Pathogenesis of diabesity-induced kidney disease: role of kidney nutrient sensing. The FEBS journal 2021, doi:10.1111/febs.15790. 
7. Cai, T.; Ke, Q.; Fang, Y.; Wen, P.; Chen, H.; Yuan, Q.; Luo, J.; Zhang, Y.; Sun, Q.; Lv, Y.; et al. Sodium-glucose cotransporter 2 inhibition suppresses HIF-1alpha-mediated metabolic switch from lipid oxidation to glycolysis in kidney tubule cells of diabetic mice. Cell Death Dis 2020, 11, 390, doi:10.1038/s41419-020-2544-7. 
8. Li, J.; Liu, H.; Takagi, S.; Nitta, K.; Kitada, M.; Srivastava, S.P.; Takagaki, Y.; Kanasaki, K.; Koya, D. Renal protective effects of empagliflozin via inhibition of EMT and aberrant glycolysis in proximal tubules. JCI insight 2020, 5, doi:10.1172/jci.insight.129034.
9. Lim, J.C.; Lim, S.K.; Han, H.J.; Park, S.H. Cannabinoid receptor 1 mediates palmitic acid-induced apoptosis via endoplasmic reticulum stress in human renal proximal tubular cells. Journal of cellular physiology 2010, 225, 654-663, doi:10.1002/jcp.22255. 
10. Drori, A.; Permyakova, A.; Hadar, R.; Udi, S.; Nemirovski, A.; Tam, J. Cannabinoid-1 receptor regulates mitochondrial dynamics and function in renal proximal tubular cells. Diabetes, obesity & metabolism 2019, 21, 146-159, doi:10.1111/dom.13497. 
11. Nam, D.H.; Lee, M.H.; Kim, J.E.; Song, H.K.; Kang, Y.S.; Lee, J.E.; Kim, H.W.; Cha, J.J.; Hyun, Y.Y.; Kim, S.H.; et al. Blockade of cannabinoid receptor 1 improves insulin resistance, lipid metabolism, and diabetic nephropathy in db/db mice. Endocrinology 2012, 153, 1387-1396, doi:10.1210/en.2011-1423. 
12. Jourdan, T.; Szanda, G.; Rosenberg, A.Z.; Tam, J.; Earley, B.J.; Godlewski, G.; Cinar, R.; Liu, Z.; Liu, J.; Ju, C.; et al. Overactive cannabinoid 1 receptor in podocytes drives type 2 diabetic nephropathy. Proceedings of the National Academy of Sciences of the United States of America 2014, 111, E5420-5428, doi:10.1073/pnas.1419901111. 

We appreciate the suggestion to add information about cannabinoids. We have included a new paragraph devoted to animal models and mechanism on p.11, lines 10-24, a paragraph about CB1 receptor inhibition on p.18, lines 8-14, as well as inclusion of CB1 in Table 1 and both figures.